# Response of Diverse Peanut Cultivars to Nano and Conventional Calcium Forms under Alkaline Sandy Soil

**DOI:** 10.3390/plants12142598

**Published:** 2023-07-09

**Authors:** Mohamed E. El-temsah, Yasser M. Abd-Elkrem, Yasser A. El-Gabry, Mohamed A. Abdelkader, Nahid A. A. Morsi, Noura M. Taha, Shaimaa H. Abd-Elrahman, Fadl A. E. Hashem, Mostafa G. Shahin, Gomaa A. Abd El-Samad, Ridha Boudiar, Cristina Silvar, Salah El-Hendawy, Elsayed Mansour, Mohamed A. Abd El-Hady

**Affiliations:** 1Agronomy Department, Faculty of Agriculture, Ain Shams University, Cairo 11241, Egyptmohamedelkader@gmail.com (M.A.A.);; 2Cell Research Department (CRD), Field Crops Research Institute (FCRI), Agricultural Research Center (ARC), Giza 12619, Egypt; nahideltemsah@agr.asu.edu.eg; 3Horticulture Department, Faculty of Agriculture, Ain Shams University, Cairo 11241, Egypt; 4Soil and Water Department, Faculty of Agriculture, Ain Shams University, Cairo 11241, Egypt; shaimaa_hassan@agr.asu.edu.eg; 5Central Laboratory for Agricultural Climate, Agricultural Research Center, Giza 12411, Egypt; 6Biotechnology Research Center—C.R.Bt Constantine, UV 03, Nouvelle Ville Ali Mendjeli, P.O. Box E73, Constantine 25016, Algeria; 7Grupo de Investigación en Bioloxía Evolutiva, CICA—Centro Interdisciplinar de Química e Bioloxía, Universidade da Coruña, 15071 A Coruña, Spain; 8Department of Plant Production, College of Food and Agriculture Sciences, King Saud University, P.O. Box 2460, Riyadh 11451, Saudi Arabia; 9Department of Crop Science, Faculty of Agriculture, Zagazig University, Zagazig 44519, Egypt

**Keywords:** nanoscale fertilizer, high pH, calcium fertilizer, peanuts yield, oil, pod, protein

## Abstract

Calcium is one of the most limiting factors for the growth and reproduction of peanut, which ultimately affects pod and seed yields. A two-year field experiment was carried out to assess the impact of five calcium applications, including nano-calcium and conventional forms, on growth, leaf nutrient content, yield traits, and quality parameters of three diverse peanut cultivars (Ismailia-1, Giza-5, and Giza-6). The applied calcium applications were calcium sulfate, which is recommended for commercial peanut cultivation and commonly referred to as gypsum (coded as Ca-1), calcium nitrate (Ca-2), nano-calcium nitrate (Ca-3), 50% calcium nitrate + 50% nano-calcium (Ca-4), and 50% calcium sulfate + 50% nano-calcium (Ca-5). Calcium sulfate (gypsum, Ca-1) was soil-supplied during the seedbed preparation as recommended, while the other calcium applications (Ca-2, Ca-3, Ca-4, and Ca-5) were exogenously sprayed three times at 30, 45, and 60 days after sowing. The soil of the experimental site was alkaline, with a high pH of 8.6. The results revealed significant differences among cultivars, calcium applications, and their interactions. The soil-supplied gypsum Ca-1 displayed lower agronomic performance on all recorded growth, leaf nutrient content, yield traits, and quality parameters. On the other hand, the foliar-supplied calcium, particularly Ca-4 and Ca-5, displayed superior effects compared to the other simple calcium forms. Ca-4 and Ca-5 produced significantly higher seed yield (3.58 and 3.38 t/ha) than the simple recommended form (Ca-1, 2.34 t/ha). This could be due to the difficulty of calcium uptake from soil-supplied calcium under high soil pH compared to the exogenously sprayed nano-calcium form. Moreover, the superior performance of Ca-4 and Ca-5 could be caused by the mixture of fertilizers from the synergistic effect of calcium and nitrate or sulfate. Furthermore, the effect of nitrate was applied in nano form in the Ca4 and Ca-5 treatments, which contributed to improving nutrient uptake efficiency and plant growth compared to the other treatments. The peanut cultivar Giza-6 showed superiority for most measured traits over the other two cultivars. The interaction effect between the assessed cultivars and calcium applications was significant for various traits. The cultivar Giza-6 showed a significant advantage for most measured traits with the mixture of 50% calcium nitrate + 50% nano-calcium (Ca-4). Conclusively, the results pointed out the advantage of the exogenously sprayed nano-calcium form combined with calcium nitrate or calcium sulfate for promoting growth, leaf nutrient content, yield, and quality traits of peanut, particularly with high-yielding cultivars under sandy soil with high pH.

## 1. Introduction

Peanut (*Arachis hypogaea* L.) is an essential and economical oleaginous crop grown in tropical and subtropical regions of the world. It offers high contents of edible oil, protein, carbohydrates, and fiber [1]. In addition, its seeds contain important minerals and vitamins, which provide high nutritive values for human consumption [2,3]. Moreover, its green canopy is used as hay for livestock [4]. Its total cultivation area is approximately 32.72 million hectares, which produces about 53.93 million tons annually [5]. However, its production must be increased due to the rapidly increasing global population and the negative impacts of global climate change on the productivity of field crops [6].

Peanut requires a considerable amount of macronutrients and micronutrients for acceptable growth and productivity [7]. Calcium is one of the most limiting factors for the growth and reproduction of peanut, which ultimately affects pod and seed yields [8]. It is required for structural roles in the cell wall and membranes. Furthermore, it is involved in signaling roles in plant growth, and it is essential for cell extension, cell division, osmoregulation, and the modulation of certain enzymes. Hence, it is a critical element in the development of pods and seeds in legume production, such as peanut and soybean [9,10]. Accordingly, calcium fertilizer improves the physiological process, plant growth, pod yield, seed quality, and storage potentiality [11,12,13,14,15,16]. 

Peanut requires higher calcium at the pod filling compared to the flowering stage. Therefore, the need for calcium is more important for the reproductive stage than for the vegetative period [17]. Most of the applied calcium (around 90%) is absorbed by pods 20–80 days after peg penetration into the soil [18]. Gypsum and lime are the commonly used calcium fertilizers, containing 23.3% and 35.5% calcium, respectively. For peanut, lime is applied at plowing, and gypsum is applied at the time of flowering [19]. Nevertheless, calcium takes a long period to be absorbed and transported to the fruit by the root system. This is inappropriate for the instantaneous demand of fruit growth and development. Moreover, calcium is most available in soil with a pH range from 7.0 to 8.0, while its availability level is usually lower under high or low soil pH. Accordingly, under these conditions, exogenous application of calcium is preferable to provide peanut plants with their requirements. The exogenous application improves calcium absorption compared to the conventional application [20]. 

The nutrient use efficiency of field crops remains low for the most important elements; thus, growers provide excessive amounts of fertilizers, which leads to environmental contamination [21,22]. Nano-technology deals with materials of tiny sizes ranging from 8 nm to 10 nm and is employed to release fertilizers of readily available nutrients. The nano-technology approach contributes to improving nutrient uptake efficiency, plant growth, and productivity [23,24,25,26,27]. Nanoparticles display improved properties based on their physical and chemical characteristics. Subsequently, nano-technology fertilizers improve food quality and have a low cost; they are also eco-friendly, and they reduce toxicity [28,29]. Several previously published reports have elucidated the importance of applied nano fertilizer in improving plant growth and productivity for various crops, such as peanut [30], apple [31], and cucumber [32]. In this context, Abdelghany et al. [33] noted that calcium in nano form improved plant height, crop growth rate, 100-seed weight, shelling percentage, seed yield, oil content, and seed protein. Moreover, nano-calcium fertilizer is used at postharvest to improve the shelf-life of fruits [32]. Accordingly, the current study aimed to assess the impact of different calcium applications, including nano-calcium and conventional forms, on growth, leaf nutrient content, yield traits, and quality parameters of three diverse peanut cultivars under sandy soil with high pH. The obtained results could provide useful information to identify the best calcium fertilizer, including the nano-calcium form or conventional calcium, based on growth, leaf nutrient content, yield, and quality traits under alkaline sandy soil.

## 2. Results

### 2.1. Growth Traits

The assessed growth traits showed considerable variations among peanut cultivars and calcium treatments, as revealed by ANOVA analyses (Table 1). The significant impact of peanut cultivars and calcium applications was detected for plant height, number of branches per plant, fresh weight per plant, and dry weight per plant in both seasons. Regarding the evaluated peanut cultivars, the highest values of growth traits were assigned for the cultivar Giza-6 across both years. Meanwhile, Giza-5 was the less vigorous cultivar, as expressed by the lowest values of traits related to growth in both seasons. Concerning calcium treatments, the mixture of 50% calcium nitrate with 50% nano-calcium (Ca-4) displayed the highest values of growth compared to the other calcium applications. The application of 50% calcium sulfate with 50% nano-calcium (Ca-5) ranked overall second after the Ca-4 treatment in terms of traits related to plant growth. Otherwise, calcium sulfate (gypsum, Ca-1) had the lowest values for those traits (Table 1). Only fresh and dry weights per plant were significantly affected by the interaction between peanut cultivars and applied calcium treatments. The interactions between Giza-6 treated with Ca-4 and Ismailia-1 treated with Ca-4 had the highest fresh and dry weight values (Figure 1). Otherwise, Giza-5 treated with Ca-1 showed the lowest values (Figure 1). 

### 2.2. Leaf Nutrient Content

The evaluated nutrient content in peanut leaves was significantly affected by peanut cultivars and calcium treatments in both seasons. Giza-6 was the richest in nutrients compared to Ismailia-1 and Giza-5. Furthermore, calcium treatments Ca-4 and Ca-5 exhibited the most favorable nutrients in peanut (Table 2). Only Fe content exhibited significant variation due to the interaction effect between assessed peanut cultivars and calcium treatments across both years (Table 2). The cultivar Giza-6 combined with Ca-5 treatment had higher Fe content than the other combinations, while Giza-5 in combination with Ca-1 displayed the lowest values of Fe content compared to the other combinations (Figure 2).

### 2.3. Yield Traits

The assessed peanut cultivars and calcium treatments significantly impacted all evaluated yield-related traits in both seasons (Table 3). The cultivar Giza-6 showed superiority over the other cultivars, Ismailia-1 and Giza-5, for all evaluated yield components. This performance was reflected in higher seed and pod yields in the first and second seasons. The calcium application Ca-4 showed a superior number of pods per plant, number of seeds per plant, 100-seed weight, pod yield, seed yield, and shelling percentage followed by Ca-5 compared to the other calcium treatments (Table 3). The interaction between the two studied factors significantly affected the number of pods per plant, pod yield, and seed yield. The combination of Giza-6 and Ca-4 resulted in higher seed yield and its components compared to other combinations (Figure 3). Otherwise, calcium treatment Ca-1 possessed the lowest values of number of pods and seed yield with the three peanut cultivars (Figure 3).

### 2.4. Content of Protein and Oil in Seeds

The assessed peanut cultivars exhibited significant differences in protein and oil content in both seasons. Giza-6 possessed a superior content of protein and oil compared to Ismailia-1 and Giza-5. Furthermore, calcium treatments showed significant differences in both seasons. Ca-5 followed by Ca-4 and Ca-3 exhibited the maximum values of protein and oil content in both seasons (Table 4). The interaction effect between assessed peanut cultivars and calcium treatments was significant across both years (Table 4). The cultivar Giza-6 cultivated under the calcium treatments Ca-4 and Ca-5 showed higher protein content, whereas Giza-5 treated with Ca-1 had the lowest value (Figure 4A). Similarly, Ca-5 combined with either Giza-5 or Giza-6 displayed the highest oil content, while Ca-1 with Ismailia-1 showed the lowest oil content (Figure 4B). 

### 2.5. Relationship among Traits and Treatments

The principal component analysis (PCA), based on the mean of the two cropping seasons, explained 89.74 % of the variation of the first two PCs, where the most variation was explained by PC1 (78.65 %) (Figure 5A). All traits showed positive correlations, and most of them contributed more to the formation of PC1. Accordingly, the assessed peanut cultivars treated with calcium applications were clearly distributed along the first component (PC1). The calcium application was the main dividing factor for PC1 regardless of peanut cultivars. Ca-4 and Ca-5 tended to be on the extremely positive side of PC1, while Ca-2 and Ca-3 were in the center, and Ca-1 was on the extremely negative side of PC1. The peanuts treated with Ca-4 and Ca-5 were positively associated with all studied traits on the positive side of PC1. This indicates the higher values of these treatments for the evaluated traits. On the opposite side, three cultivars treated with Ca-1 were located on the negative side of the PC1 with low performance. Otherwise, the peanut cultivars were distributed along the PC2. Giza-5 was revealed on the top of PC2, followed by Giza-6, while Ismailia-1 was found on the negative side of PC2 (Figure 5A). Likewise, the heatmap and hierarchical clustering based on the evaluated growth, yield, and quality traits separated the assessed peanut cultivars and calcium applications into different clusters (Figure 5B). The combination effect between Giza-6 and Ca-4 and C-5 displayed superior values for all studied traits (represented in blue). In contrast, the three cultivars treated with Ca-1 possessed the lowest values (red values).

## 3. Discussion

Fertilizer management plays an essential role in boosting peanut productivity alongside maintaining soil fertility in the long term. Calcium fertilization is irreplaceable in peanut cultivation due to its involvement in peg development as well as fruit formation and maturity [34,35,36]. Consequently, an insufficient amount of calcium causes anomalies in pod and seed formation, which results in empty pods and inadequately filled pods [10,37]. In the present study, the impact of different calcium treatments, including nano or conventional forms, in newly reclaimed sandy soil on peanut growth, leaf nutrient content, yield, and quality was assessed. The applied calcium fertilizers, including the nano-calcium form combined with calcium nitrate (Ca-4) and with calcium sulfate (Ca-5), displayed a considerably superior impact on peanut growth, leaf nutrient content, yield traits, and better quality compared to the supplied fertilizer separately (Ca-1, Ca-2, and Ca-3). The recommended application of gypsum (calcium sulfate, Ca-1) was supplied directly into the soil, and it was in direct contact with the fruiting zone to provide peanut fruits with delivered calcium (Ca^2+^). Nonetheless, the soil-supplied Ca-1 exhibited a lower agronomic performance compared to the other applications. This trial was performed in soil with high pH (8.6). The most favorable pH range for calcium availability and crop growth is between 7 and 8 [38]. This could be a reason for the lower agronomic impact of soil-supplied gypsum (Ca-1). Moreover, the difficulty of calcium uptake from the soil and its distribution through phloem in the plant compared to the exogenously sprayed nano-calcium form could impact that result [9]. The obtained results regarding soil-supplied gypsum are inconsistent with Hamza et al. [34], who disclosed a highly positive impact of supplied gypsum into the soil combined with Ca(NO_3_)_2_ on seed yield, oil content, and protein content. Otherwise, our findings are in consonance with other studies [36,39], which found that exogenously applied calcium was more effective in avoiding calcium deficiency in peanuts compared to calcium supplied into the soil. 

In addition, the superior performance under treatments of Ca-4 (nano-calcium combined with calcium nitrate) and Ca-5 (nano-calcium combined with calcium sulfate) could be caused by the mixture of fertilizers from the synergistic effect of calcium and nitrate or sulfate. Furthermore, the effect of nitrate was applied in nano form in the Ca4 and Ca-5 treatments, which contributed to improving nutrient uptake efficiency and plant growth compared to the other treatments. This was clear in the treatment of nano-calcium nitrate (Ca-3), which had overall higher values of all traits compared to the conventional calcium nitrate fertilizer (Ca-2), which reinforced the abovementioned results. Moreover, calcium sulfate is a rich source of sulfur. Sulfur is a major plant nutrient; it plays a vital role in chlorophyll synthesis, and it assimilates partitioning. Furthermore, it enhances flowering, fruiting, and leaves, resulting in better growth and yield components. Additionally, it is required for the synthesis of oil, protein, and vitamins. Accordingly, the exogenous application of calcium nanoparticles (in particular, Ca-4 and Ca-5) prevented calcium deficiency disorders under sandy soil conditions and enhanced peanut productivity and quality compared to the other treatments. Furthermore, it is worth noting that Fe content was higher in Ca-4 and Ca-5 compared to the recommended application for commercial peanut cultivation under sandy soils with calcium sulfate Ca-1. The applications of Ca-4 and Ca-5 increased the total volume of plant root hairs, which in turn increased the siderophores in the rhizosphere, which could increase nutrient availability in the soil, including Fe. Moreover, applied calcium through the foliar application was taken up directly by leaves without interacting with Fe available in the soil. Otherwise, applying gypsum into the soil in Ca-1 could increase the concentration of bicarbonate ions and restrict the translocation of iron to the interveinal cells of the upper part of the branches [40]. In this context, Xiumei et al. [30] noted that nano-Ca(NO_3_)_2_ combined with organic fertilizers improved the physiological response of the peanut and its nutrient uptake compared to conventional forms. Similarly, Janmohammadi et al. [41], Salama [42], and Zheng et al. [43] deduced that nano fertilizers provide readily available nutrients and improved overall plant growth and development.

The evaluated peanut cultivars showed considerable variability in all measured traits under different calcium treatments. Giza-6 exhibited superiority for most measured growth, leaf nutrient content, yield, and quality traits over the other two peanut cultivars (Ismailia-1 and Giza-5) under different calcium applications. This indicates the genetic variability of peanut cultivars in their response to different calcium applications. Likewise, previously published studies elucidated that the peanut genotypes behaved differently in their response to calcium treatments. In this context, Ahmed and Zeidan [44] found significant differences in pod and seed yield as well as yield attributes among peanut cultivars under calcium treatments. Moreover, Abdalla et al. [4] and Abd-El-Motaleb and Yousef [45] disclosed that peanut cultivars with erect growth habits (Giza-5) outperformed semi-erect cultivars (Giza-4) regarding plant height, 100-seed weight, and pod yield. The interaction effect between peanut cultivars and calcium treatment was significant for different traits. The cultivar Giza-6 treated with calcium nitrate or calcium sulfate combined with nano-calcium outperformed the other cultivars under different calcium treatments. This outperformance was observed for growth, leaf nutrient content, yield, and quality-related traits. Therefore, the response of peanut cultivars is dependent on agricultural management, including calcium fertilizers. Thus, high-yielding peanut cultivars, such as Giza-6, combined with a mixture of nano-calcium with calcium nitrate or calcium sulfate are highly recommended to enhance peanut productivity and quality, particularly under alkaline sandy soil conditions. Based on the multivariate analysis, principal component biplot, and heatmap and hierarchical clustering, all measured traits were positively correlated. This indicates that the overall vigor of plant growth led to higher nutrient uptake and to improved seed yield and quality. Accordingly, cultivating advantageous genotypes with vigorous growth and high leaf nutrient content could assist in producing high-yielding peanuts with increased oil and protein percentages under recommended agricultural practices, including calcium fertilizers. In this context, Gomes and Lopes [46] found that the number and weight of pods, number and weight of mature seeds, shelling percentage, 100-seed weight, number of primary and secondary branches per plant, and harvest index were positively associated with pod and seed yield. 

## 4. Materials and Methods

### 4.1. Experimental Site Conditions

A field experiment was carried out for two summer seasons in 2020 and 2021 in newly reclaimed soil at the Al-Mansouriya farm, Giza, Egypt (30°07′35.2″ N, 31°04′14.9″ E). Soil samples were taken before the experiment setup and analyzed [47]. The analysis revealed that the soil was sandy throughout the profile (63.0% sand, 27.9% silt, and 9.10% clay). Organic matter, calcium carbonate, electrical conductivity, and pH were 0.85%, 2.4%, 1.16 dS/m, and 8.6, respectively. In addition, the available N, P, and K were 12.0, 8.0, and 98 µg/g dry soil, respectively. The experimental site was characterized by high temperatures and no rainfall during the summer seasons. Minimum and maximum temperatures and relative humidity were overall similar across the two cropping seasons (Table 5).

### 4.2. Experimental Design and Cultural Management

The experimental design was laid out in a randomized complete block design with split-plot arrangement in three replications. Three peanut cultivars (Giza-5, Giza-6, and Ismailia-1) were assigned to the main plots, while the subplots were allocated to five forms of calcium fertilizers: calcium sulfate, which is recommended for commercial peanut cultivation and commonly referred to as gypsum (CaSO_4_. 2H_2_O; 19.5% Ca), was coded as Ca-1, calcium nitrate (Ca(NO_3_)_2_. 4H_2_O; 23% Ca) was coded as Ca-2, nano-calcium nitrate was coded as Ca-3, 50% calcium nitrate + 50% nano-calcium nitrate was coded as Ca-4, and 50% calcium sulfate + 50% nano-calcium nitrate was coded as Ca-5. Before sowing, peanut seeds were inoculated with specific Rhizobium bacteria species (*Rhizobium leguminosarum*). Sowing was performed by hand on hills in the first week of May for both cropping seasons (2020 and 2021).

Each plot included five ridges; each was 4 m long and 60 cm apart (and 20 cm among hills), resulting in a plot area of 12 m^2^ (3 m wide and 4 m long). After sowing, the seeds were immediately irrigated, and after three weeks, the seedlings were thinned to two seedlings per hill to obtain a planting density of 167,000 plants ha^−1^. Hence, each plot included 200 peanut plants. Superphosphate (15.5% P_2_O_5_) was incorporated into the soil during the bed seed preparation at a rate of 360 kg ha^−1^, while potassium sulfate (K_2_SO_4_ contains 48% K_2_O) was delivered directly before flowering (240 potassium sulfate kg ha^−1^). Nitrogen fertilizer was added in urea form (46%) at a rate of 150 kg ha^−1^ fractioned into two equal doses. The first N supply was delivered at sowing, and the second one was provided 15 days after sowing. Gypsum treatment was ground and incorporated into the soil at a rate of 3.5 tons ha^−1^ during the seedbed preparation. Both calcium nitrate and nano-calcium were used as foliar applications at a rate of 715 mg ha^−1^ each, divided into three application rates at 30, 45, and 60 days after sowing. Nano-calcium (calcium nitrate nanoparticles) was applied at a rate of 3 g/L (1440 g/ha), divided into three applications at 30, 45, and 60 days after sowing. The foliar applications were applied in the morning time (08:00–10:00 a.m.) on a sunny and dry day. Irrigation water was applied using a drip irrigation system with an emitter discharge of 4.0 L h^−1^ at an operating pressure of 1.0 bar. Irrigation was applied daily until plant emergence, and then plants were watered overall at 7-day intervals depending on weather conditions.

### 4.3. Preparation of Calcium Nitrate Nanoparticles

The preparation process was carried out in the laboratory of the Genetic Engineering Department, Faculty of Agriculture, Ain Shams, Egypt. The calcium nitrate nanoparticles were obtained by placing calcium nitrate, Ca(NO_3_)_2_, in portions of 80 g into four stainless steel canisters with large, medium, and small metal balls. Then, the canisters were moved to the ball milling machine and shacked for 30 h at 1000 rpm/minute. Finally, the size of the milled Ca(NO_3_)_2_ particles was determined by transmission electron microscopy (TEM, Jeol-1400, Tokyo, Japan) (Figure 6) at the Metallurgical Research Institute, Egypt. 

### 4.4. Recorded Traits

Traits were recorded at harvest on 12 and 17 September in the first and second seasons, respectively. The following traits were measured based on ten random plants selected from three inner ridges in each plot: plant height (cm), number of pods per plant, number of seeds per plant, fresh weight per plant (g), and dry weight per plant (g). The remaining peanut plants in the plots were harvested and sun dried to reach 15% moisture content. After that, the following traits were assessed: pod yield (kg/ha), seed yield (kg/ha), 100-seed weight (g), shelling percentage (%), protein percentage (%), and oil percentage (%). Total nitrogen was determined in seeds using the Kjeldahl method, and protein content was calculated by multiplying the total nitrogen by 6.25 [48]. The oil percentage was determined after extraction by the Soxhlet extraction method. Samples of leaves were oven dried at 70 °C until constant weight and used to determine the content of total nitrogen (N%), phosphorus (P%), and potassium (K%) according to the methods described by Chapman and Pratt [49], Watanabe and Olsen [50], and Chapman and Pratt [49], respectively. Total micro and heavy elements were determined using ICP Mass Spectrometry [51].

### 4.5. Statistical Analysis

Data analyses were performed using R software version 4.2.1. ANOVA analyses were performed for each year separately. Differences among peanut cultivars, calcium fertilizers, and their interactions were separated by Tukey’s HSD test (*p* < 0.05). Principal component analysis and heatmap and hierarchical clustering were performed using FactoMineR [52] and pheatmap packages [53], respectively, implemented in R software.

## 5. Conclusions

Calcium fertilization is essential for improving peanut growth, productivity, and quality. The impact of different calcium treatments, including nano or conventional forms, was assessed on diverse peanut cultivars in newly reclaimed sandy soil. The soil-supplied calcium sulfate (gypsum, Ca-1) exhibited lower agronomic performance compared to the other calcium applications. This could be attributed to the high soil pH (8.6) of the experimental site, which lowered the calcium availability level. Furthermore, this could be attributable to the difficulty of calcium uptake from the soil and its distribution through phloem in the plant compared to the exogenously sprayed nano-calcium form. In addition, the positive impact of Ca-4 (nano-calcium combined with calcium nitrate) and Ca-5 (nano-calcium combined with calcium sulfate) on peanut productivity could be caused by the mixture of fertilizers from the synergistic effect of calcium and nitrate or sulfate. Moreover, nanoparticles contributed to improving nutrient uptake efficiency, plant growth, and productivity. Accordingly, the exogenously sprayed calcium nitrate or calcium sulfate combined with nano-calcium is more suitable for high-yielding peanut cultivars, particularly under high soil pH, to improve peanut productivity and quality. 

## Figures and Tables

**Figure 1 plants-12-02598-f001:**
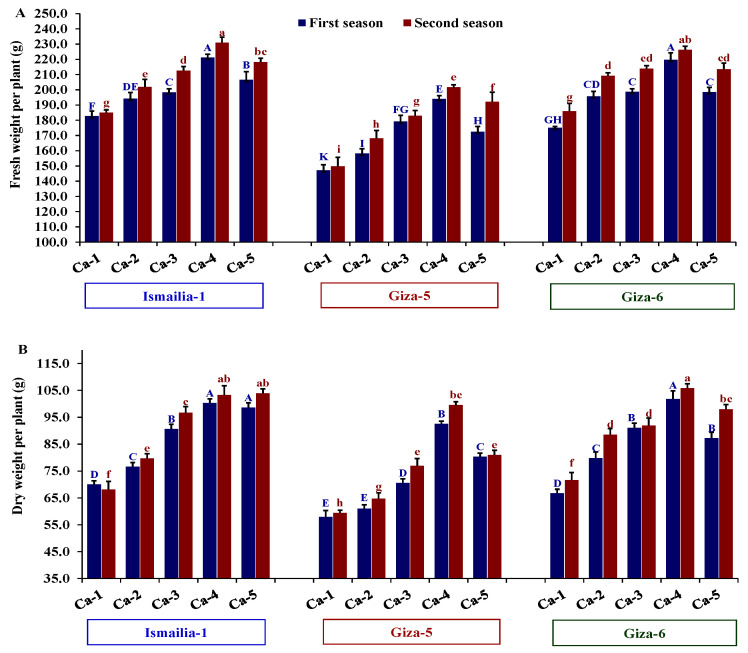
Influence of different calcium application forms (nano-calcium and conventional) on fresh weight per plant (**A**) and dry weight per plant (**B**) of three peanut cultivars in the 2020 and 2021 growing seasons. Ca-1: gypsum calcium sulfate, Ca-2: calcium nitrate, Ca-3: nano-calcium, Ca-4: 50% calcium nitrate + 50% nano-calcium, Ca-5: 50% calcium sulfate + 50% nano-calcium. The bars on the tops of the columns represent the SE, and different letters on the columns show the significant difference using Tukey’s HSD test (*p* < 0.05). The uppercase letters belong to the first season, while the lowercase letters belong to the second season.

**Figure 2 plants-12-02598-f002:**
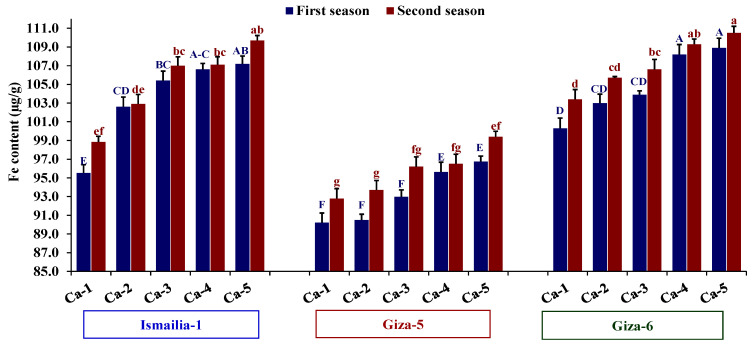
Influence of different calcium application forms (nano-calcium and conventional) on Fe content in three peanut cultivars in the 2020 and 2021 growing seasons. Ca-1: gypsum calcium sulfate, Ca-2: calcium nitrate, Ca-3: nano-calcium, Ca-4: 50% calcium nitrate + 50% nano-calcium, Ca-5: 50% calcium sulfate + 50% nano-calcium. The bars on the tops of the columns represent the SE, and different letters on the columns show the significant difference using Tukey’s HSD test (*p* < 0.05). The uppercase letters belong to the first season, while the lowercase letters belong to the second season.

**Figure 3 plants-12-02598-f003:**
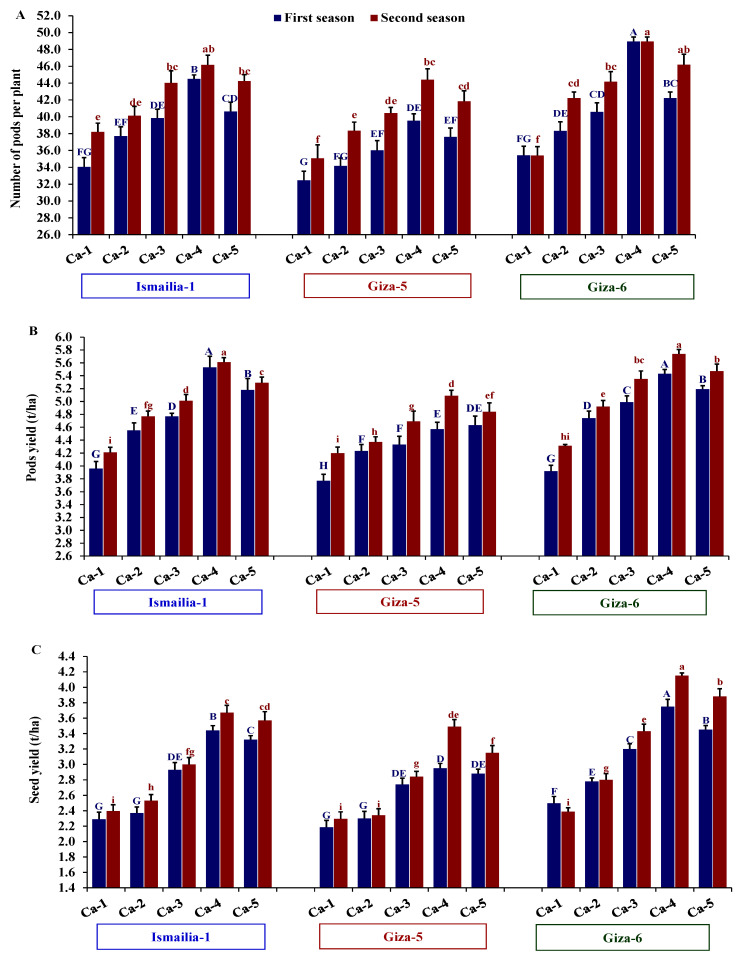
Influence of different calcium application forms (nano-calcium and conventional) on number of pods per plant (**A**), pod yield (**B**), and seed yield (**C**) of three peanut cultivars in the 2020 and 2021 growing seasons. Ca-1: gypsum calcium sulfate, Ca-2: calcium nitrate, Ca-3: nano-calcium, Ca-4: 50% calcium nitrate + 50% nano-calcium, Ca-5: 50% calcium sulfate + 50% nano-calcium. The bars on the tops of the columns represent the SE, and different letters on the columns show the significant difference using Tukey’s HSD test (*p* < 0.05). The uppercase letters belong to the first season, while the lowercase letters belong to the second season.

**Figure 4 plants-12-02598-f004:**
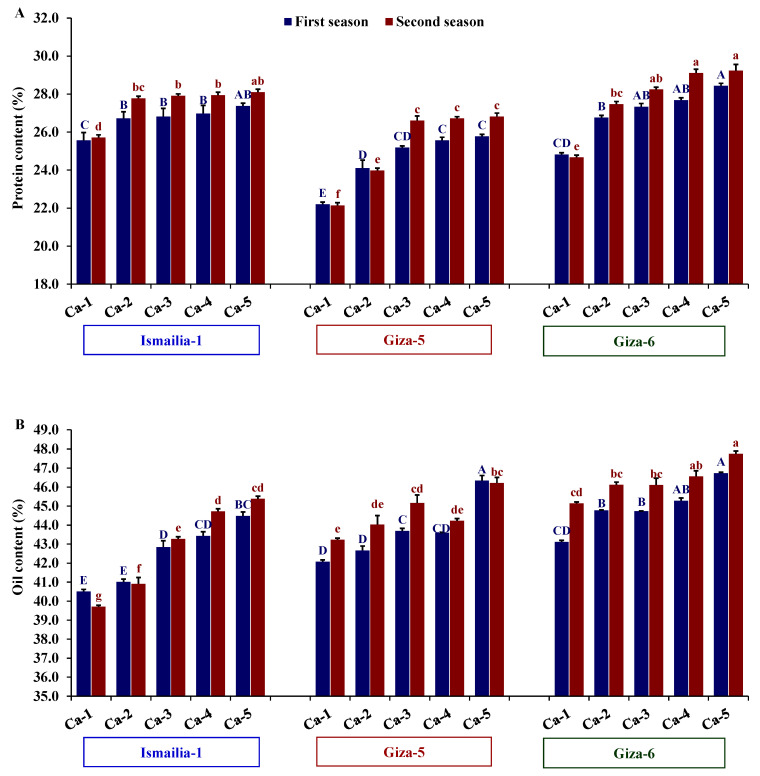
Influence of different calcium application forms (nano-calcium and conventional) on protein content (**A**) and oil content (**B**) in three peanut cultivars in the 2020 and 2021 growing seasons. Ca-1: gypsum calcium sulfate, Ca-2: calcium nitrate, Ca-3: nano-calcium, Ca-4: 50% calcium nitrate + 50% nano-calcium, Ca-5: 50% calcium sulfate + 50% nano-calcium. The bars on the tops of the columns represent the SE, and different letters on the columns show the significant difference using Tukey’s HSD test (*p* < 0.05). The uppercase letters belong to the first season, while the lowercase letters belong to the second season.

**Figure 5 plants-12-02598-f005:**
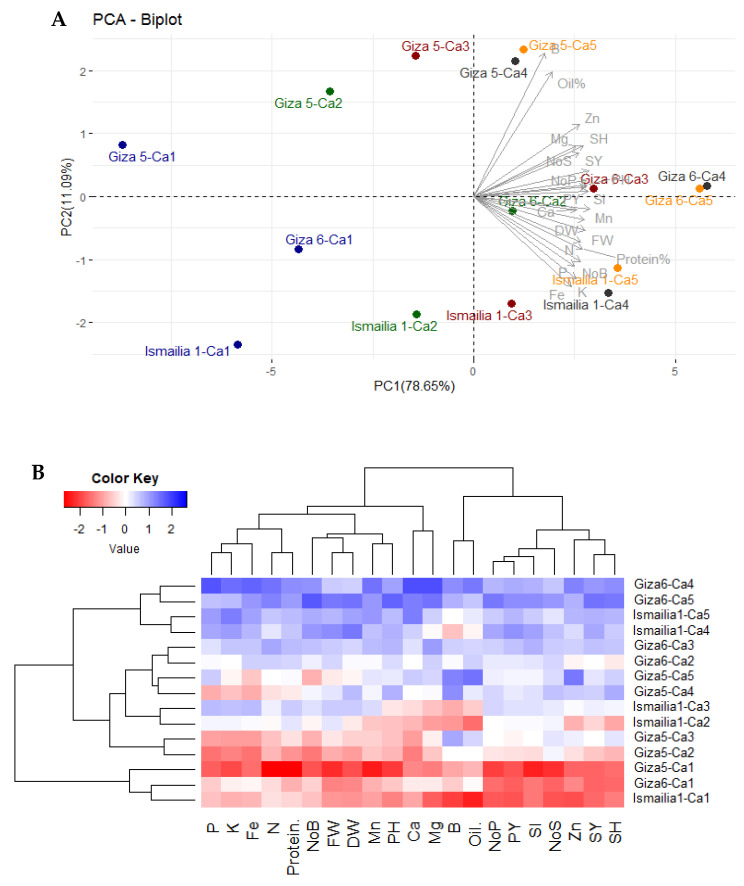
Principal component biplot (**A**) and heatmap and hierarchical clustering (**B**) for the assessed peanut cultivars and calcium applications. Calcium applications are characterized by colors in the PC biplot. In the heatmap, red and blue colors indicate high and low values for the corresponding trait, in the same order. Oil%: oil percentage, NoP: number of pods per plant, SH: shelling percentage, NoS: number of seeds per plant, PH: plant height, FW: fresh weight per plant, DW: dry weight per plant, PY: pod yield, SY: seed yield, Protein%: protein content, N: nitrogen content, P: phosphorus content, K: potassium content, Ca: calcium content, Mg: magnesium content, Fe: iron content, Mn: manganese content, Zn: zinc content, B: boron content.

**Figure 6 plants-12-02598-f006:**
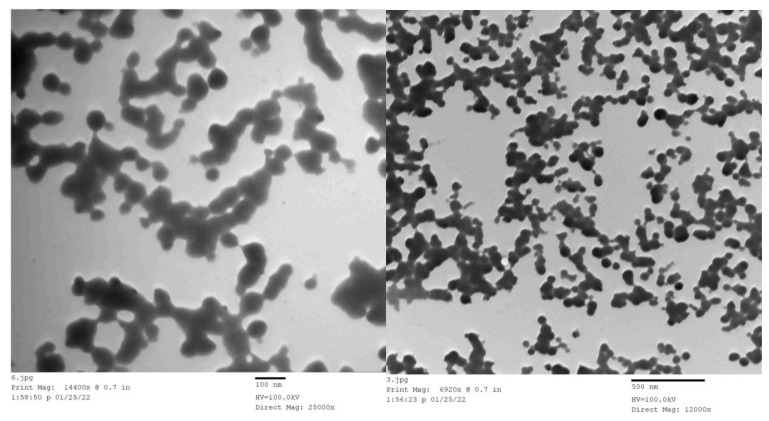
Transmission electron microscopy (TEM) of nano-calcium particles using 25,000× (**left**) and 12,000× (**right**) magnification.

**Table 1 plants-12-02598-t001:** Influence of different calcium application forms (nano-calcium and conventional) on growth traits of three peanut cultivars in the 2020 and 2021 growing seasons.

Studied Factor		Plant Height (cm)		No. of Branches per Plant		Fresh Weight per Plant (g)		Dry Weight per Plant (g)	
		2020	2021	2020	2021	2020	2021	2020	2021
Peanut cultivar (G)									
Ismailia-1		43.86 b	47.91 b	10.26 a	10.54 a	200.58 a	209.70 a	87.21 a	90.33 a
Giza-5		42.48 b	47.01 b	9.06 b	9.33 b	170.14 c	178.90 b	72.48 b	76.33 b
Giza-6		47.27 a	51.15 a	10.4 a	11.07 a	197.52 b	209.74 a	85.34 a	91.19 a
Calcium fertilization (Ca)									
Ca-1		40.39 d	44.79 e	9.02 c	9.28 c	168.27 d	173.57 e	64.88 e	66.42 e
Ca-2		42.61 c	46.88 d	9.41 c	9.91 b	182.75 c	193.03 d	72.48 d	77.66 d
Ca-3		43.25 c	48.40 c	10.12 b	10.22 b	192.16 b	203.15 c	84.09 c	88.51 c
Ca-4		49.28 a	52.51 a	10.82 a	11.3 a	211.59 a	225.3 a	100.93 a	109.57 a
Ca-5		47.14 b	50.87 b	10.18 b	10.84 a	192.48 b	208.02 b	88.70 b	94.30 b
ANOVA	df	*p*-value							
G	2	<0.001	<0.001	<0.001	<0.001	<0.001	<0.001	<0.001	<0.001
Ca	4	<0.001	<0.001	<0.001	<0.001	<0.001	<0.001	<0.001	<0.001
G × Ca	8	0.325	0.061	0.059	0.807	0.002	0.011	<0.001	<0.001

The presented averages correspond to the main effect of studied factors (peanut cultivars and calcium fertilization) on evaluated traits. Ca-1: gypsum calcium sulfate, Ca-2: calcium nitrate, Ca-3: nano-calcium, Ca-4: 50% calcium nitrate + 50% nano-calcium, Ca-5: 50% calcium sulfate + 50% nano-calcium. Means followed by different letters under the same factor were significantly different according to Tukey’s HSD test (*p* < 0.05).

**Table 2 plants-12-02598-t002:** Influence of different calcium application forms (nano-calcium and conventional) on leaf nutrient content of three peanut cultivars in the 2020 and 2021 growing seasons.

Studied Factor		N (%)		P (%)		K (%)		Ca (%)		Mg (%)	
		2020	2021	2020	2021	2020	2021	2020	2021	2020	2021
Peanut cultivar (G)											
Ismailia-1		4.21 b	4.39 b	0.82 a	0.86 a	0.63 a	0.67 a	1.71 b	1.77 b	0.21 c	0.21 c
Giza-5		3.87 c	4.05 c	0.72 b	0.73 b	0.55 b	0.56 b	1.54 c	1.65 c	0.22 b	0.23 b
Giza-6		4.38 a	4.63 a	0.82 a	0.87 a	0.65 a	0.66 a	1.87 a	1.97 a	0.28 a	0.29 a
Calcium fertilization (Ca)											
Ca-1		3.71 c	4.04 c	0.71 d	0.73 d	0.56 d	0.56 d	1.51 d	1.60 d	0.18 e	0.19 e
Ca-2		4.17 b	4.26b c	0.74 c	0.79 c	0.58 c	0.61 c	1.52 d	1.63 d	0.21 d	0.23 d
Ca-3		4.2 b	4.41 ab	0.79 b	0.83 bc	0.62 b	0.63 bc	1.62 c	1.71 c	0.24 c	0.25 c
Ca-4		4.31 ab	4.48 ab	0.82 b	0.84 b	0.64 b	0.66 b	1.87 b	1.94 b	0.27 b	0.27 b
Ca-5		4.38 a	4.58 a	0.89 a	0.92 a	0.66 a	0.71 a	2.02 a	2.11 a	0.29 a	0.29 a
ANOVA	df	*p*-value									
G	2	<0.001	<0.001	<0.001	<0.001	<0.001	<0.001	<0.001	<0.001	<0.001	<0.001
Ca	4	<0.001	<0.001	<0.001	<0.001	<0.001	<0.001	<0.001	<0.001	<0.001	<0.001
G × Ca	8	0.064	0.648	0.185	0.079	0.265	0.484	0.611	0.102	0.392	0.494
**Studied** **factor**		**Fe (µg/g)**		**Mn (µg/g)**		**Zn (µg/g)**		**B (µg/g)**			
		**2020**	**2021**	**2020**	**2021**	**2020**	**2021**	**2020**	**2021**		
Peanut cultivar (G)											
Ismailia-1		103.47 a	105.11 a	26.14 a	28.51 b	42.63 c	45.1 c	1.46 b	1.60 c		
Giza-5		93.21 b	95.72 b	24.39 b	26.33 c	45.5 b	47.23 b	1.69 a	1.86 a		
Giza-6		104.86 a	107.10 a	26.9 a	29.81 a	47.34 a	48.81 a	1.72 a	1.8.00 b		
Calcium fertilization (Ca)											
Ca-1		95.35 d	98.34 d	22.62 d	23.84 d	38.27 e	39.91 e	1.43 d	1.55 e		
Ca-2		98.70 c	100.77 c	24.42 c	27.52 c	42.2 d	43.16 d	1.57 c	1.70 d		
Ca-3		100.76 b	103.27 b	26.12 b	29.02 b	46.64 c	49.61 c	1.67 b	1.77 c		
Ca-4		103.48 a	104.30 b	27.42 ab	30.05 ab	49.37 b	52.39 b	1.69 b	1.84 b		
Ca-5		104.28 a	106.53 a	28.48 a	30.64 a	54.3 a	56.16 a	1.76 a	1.90 a		
ANOVA	df	*p*-value									
G	2	<0.001	<0.001	<0.001	<0.001	<0.001	<0.001	<0.001	<0.001		
Ca	4	<0.001	<0.001	<0.001	<0.001	<0.001	<0.001	<0.001	<0.001		
G × Ca	8	0.006	0.026	0.811	0.138	0.916	0.058	0.163	0.058		

The presented averages correspond to the main effect of studied factors (peanut cultivars and calcium fertilization) on evaluated traits. Ca-1: gypsum calcium sulfate, Ca-2: calcium nitrate, Ca-3: nano-calcium, Ca-4: 50% calcium nitrate + 50% nano-calcium, Ca-5: 50% calcium sulfate + 50% nano-calcium. Means followed by different letters under the same factor were significantly different according to Tukey’s HSD test (*p* < 0.05).

**Table 3 plants-12-02598-t003:** Influence of different calcium application forms (nano-calcium and conventional) on yield traits of three peanut cultivars in the 2020 and 2021 growing seasons.

Studied Factor		No. of Pods per Plant		No. of Seeds per Plant		100-Seed Weight (g)		Pods Yield (t/ha)		Seed Yield (t/ha)		Shelling (%)	
		2020	2021	2020	2021			2020	2021	2020	2021	2020	2021
Peanut cultivar (G)													
Ismailia-1		39.35 b	42.56 a	60.18 b	63.58 b	57.80 b	61.52 b	4.80 a	4.98 b	2.87 b	3.03 b	59.82 b	60.93 b
Giza-5		35.96 c	40.02 b	63.05 a	60.59 c	52.45 c	55.26 c	4.31 b	4.64 c	2.61 c	2.82 c	60.64 b	60.87 b
Giza-6		41.10 a	43.38 a	62.48 a	67.10 a	60.31 a	63.98 a	4.85 a	5.16 a	3.14 a	3.33 a	64.59 a	64.55 a
Calcium fertilization (Ca)													
Ca-1		33.98 d	36.22 d	52.85 d	53.65 c	46.89 d	52.74 d	3.88 e	4.24 e	2.32 e	2.36 e	59.83 b	55.64 c
Ca-2		36.74 c	40.22 c	61.58 c	64.34 b	54.54 c	60.16 c	4.51 d	4.69 d	2.48 d	2.56 d	55.10 c	54.55 c
Ca-3		38.80 b	42.88 b	63.38 bc	65.18 b	59.33 b	64.24 b	4.70 c	5.02 c	2.96 c	3.09 c	62.95 a	61.59 b
Ca-4		44.32 a	46.51 a	67.72 a	69.67 a	63.63 a	69.45 a	5.18 a	5.48 a	3.38 a	3.77 a	65.29 a	68.80 a
Ca-5		40.15 b	44.09 b	64.00 b	65.92 b	59.87 b	66.01 b	5.00 b	5.20 b	3.22 b	3.53 b	64.33 a	67.95 a
ANOVA	df				*p*-value								
G	2	<0.001	<0.001	0.008	<0.001	<0.001	<0.001	<0.001	<0.001	<0.001	<0.001	0.032	0.044
Ca	4	<0.001	<0.001	<0.001	<0.001	<0.001	<0.001	<0.001	<0.001	<0.001	<0.001	<0.001	<0.001
G × Ca	8	0.007	0.042	0.498	0.661	0.054	0.129	<0.001	0.003	0.014	0.029	0.675	0.688

The presented averages correspond to the main effect of studied factors (peanut cultivars and calcium fertilization) on evaluated traits. Ca-1: gypsum (calcium sulfate), Ca-2: calcium nitrate, Ca-3: nano-calcium, Ca-4: 50% calcium nitrate + 50% nano-calcium, Ca-5: 50% calcium sulfate + 50% nano-calcium. Means followed by different letters under the same factor were significantly different according to Tukey’s HSD test (*p* < 0.05).

**Table 4 plants-12-02598-t004:** Influence of different calcium application forms (nano-calcium and conventional) on protein and oil contents of three peanut cultivars in the 2020 and 2021 growing seasons.

Studied Factor		Protein Content (%)		Oil Content (%)	
		2020	2021	2020	2021
Peanut cultivar (G)					
Ismailia-1		26.69 b	27.34 a	42.46 c	42.80 b
Giza-5		24.16 c	25.25 c	43.67 b	44.57 a
Giza-6		27.01 a	27.74 a	44.92 a	46.34 a
Calcium fertilization (Ca)					
Ca-1		24.19 d	24.17 e	41.90 d	42.69 d
Ca-2		25.86 c	26.74 d	42.81 cd	43.69 cd
Ca-3		26.44 b	28.25 c	43.75 bc	44.85 bc
Ca-4		26.74 ab	28.59 b	44.11 ab	45.17 ab
Ca-5		27.19 a	29.05 a	45.85 a	46.45 a
ANOVA	df	*p*-value			
G	2	<0.001	<0.001	<0.001	<0.001
Ca	4	<0.001	<0.001	<0.001	<0.001
G × Ca	8	<0.001	<0.001	<0.001	0.036

The presented averages correspond to the main effect of studied factors (peanut cultivars and calcium fertilization) on evaluated traits. Ca-1: gypsum calcium sulfate, Ca-2: calcium nitrate, Ca-3: nano-calcium, Ca-4: 50% calcium nitrate + 50% nano-calcium, Ca-5: 50% calcium sulfate + 50% nano-calcium, N: nitrogen content, P: phosphorus content, K: potassium content, Ca: calcium content, Mg: magnesium content, Fe: iron content, Mn: manganese content, Zn: zinc content, B: boron content. Means followed by different letters under the same factor were significantly different according to Tukey’s HSD test (*p* < 0.05).

**Table 5 plants-12-02598-t005:** Climatic data of the experimental site during the 2020 and 2021 seasons.

Month	Maximum Temperature °C	Minimum Temperature °C	Average Relative Humidity %	Solar Radiation (MJ/m^2^ day^−1^)	Wind Speed m/s	Rainfall (mm)
First season (2020)						
May	35.0	15.5	33.7	21.0	1.16	0.00
June	35.0	19.1	39.3	23.3	1.92	0.00
July	35.8	20.2	42.1	24.0	1.85	0.00
August	35.0	20.8	43.3	22.3	2.19	0.00
September	32.6	18.8	46.9	19.7	1.71	0.00
Second season (2021)						
May	32.5	13.5	38.3	23.1	1.43	0.00
June	34.0	17.8	39.4	25.2	1.41	0.00
July	36.3	19.9	41.2	25.7	1.91	0.00
August	36.2	20.0	43.0	22.1	2.22	0.00
September	34.7	19.3	46.5	19.5	1.62	0.00

## Data Availability

The data presented in this study are available upon request from the corresponding author.

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
