# Peer review of "Response of Diverse Peanut Cultivars to Nano and Conventional Calcium Forms under Alkaline Sandy Soil"

_plants, 2023, doi:10.3390/plants12142598_

Round 1
Reviewer 1 Report
Studies like presented „Response of Diverse Peanut Cultivars to Nano and
Conventional Calcium Forms Under Alkaline Sandy Soil“ are very necessary, time-consuming and organizationally demanding.
The authors have done this brilliantly, but I have the following comments on the presentation of the introduction, results and discussion.
There should be more information about nano-calcium in the Introduction section; e.g.: What is the oxidation number of calcium in nano-particles? What plants has nano-calcium been used on before? When was nanocalcium developed?
All the tables (1-4) are confusing because only one set of Ca1-Ca5 treatments is listed, but which cultivar does it belong to? Ismalia-1, Giza-5 or Giza 6? And also statistics, which cultivar does it belong to? All cultivars were treated with CA1-CA5. Comparing Table 1 with Fig. 1 I guess that it is cultivar Giza-6. However in this case fresh weight and dry weight per plant of cultivar Giza-6 are presented twice, firstly in table 1 and secondly in Fig. 1. Similarly, number of pods per plant, plant yield and seed yield is presented twice, firstly in Table 3 and also in Fig. 3 and similarly Table 4 and Figure 4 with protein and oil content. The results should be presented only once.
Moreover, in all Figures non-treated control (without additional Ca) should be also involved. I think that it is key for comparison.
In all figures, only 1 decimal place on the y-axis would be sufficient.
Increased content of Fe in plants treated with 50% calcium sulfate + 50% nano-calcium is not discussed. Can it be related to the synthesis of siderophores? In addition, in discussion section the effect of sulfate alone is not mentioned thoroughly.
Line 210: Zinc instead of Zink
B: boron instead of bron
The last sentence of the discussion (line 279-282) makes no sense, Ca is not mentioned.
Methods: How many peanut plants did you have in total? How many controls and how many in each group (Ca1-Ca5)?
What was the commercial source of nano-calcium nitrate? (producer, country of origin, batch)
Line 290 Please express the conductivity in Siemens
Line 312 potassium sulfate is not K2O
Line 319 is confusing: at a rate 3g.l-1 (1440 g/ha), Why two different concentrations? Which is the stock solution?
Line 341: Which methods were used for protein and lipid content determination?
Line 360: Why Calcium C in capitals?
Writing of references should be uniform, e.g. journal abbreviations and full titles are mixed.
Minor editing of English language required
Author Response
Dear Editor,
We would like to thank you and the reviewers for the time and efforts devoted to our manuscript entitled “Response of Diverse Peanut Cultivars to Nano and Conventional Calcium Forms Under Alkaline Sandy Soil” (plants-2421890). We have revised the manuscript according to the comments and suggestions pointed out by the reviewers. We have addressed the comments of the reviewers in a point-by-point below in red color; in addition, we have highlighted all the associated changes made to the manuscript using track changes.
Yours sincerely,
Authors
Responses to Reviewers' Comments
Reviewer 1:
Studies like presented “Response of Diverse Peanut Cultivars to Nano and Conventional Calcium Forms Under Alkaline Sandy Soil” are very necessary, time-consuming and organizationally demanding. The authors have done this brilliantly, but I have the following comments on the presentation of the introduction, results and discussion.
Re: We would like to thank the Reviewer for his time dedicated to our manuscript and his positive assessment of our work.
There should be more information about nano-calcium in the Introduction section; e.g.: What is the oxidation number of calcium in nano-particles? What plants has nano-calcium been used on before? When was nano calcium developed?
Re: More information has been added as suggested, please see lines 65-70 and 88-99
All the tables (1-4) are confusing because only one set of Ca1-Ca5 treatments is listed, but which cultivar does it belong to? Ismalia-1, Giza-5 or Giza 6? And also statistics, which cultivar does it belong to? All cultivars were treated with CA1-CA5. Comparing Table 1 with Fig. 1 I guess that it is cultivar Giza-6. However, in this case fresh weight and dry weight per plant of cultivar Giza-6 are presented twice, firstly in table 1 and secondly in Fig. 1. Similarly, number of pods per plant, plant yield and seed yield is presented twice, firstly in Table 3 and also in Fig. 3 and similarly Table 4 and Figure 4 with protein and oil content. The results should be presented only once.
Moreover, in all Figures non-treated control (without additional Ca) should be also involved. I think that it is key for comparison.
Re: Tables 1-4 presented the mean effects of studied factors calcium fertilization (Ca1 - Ca5) and peanut cultivars (Ismalia-1, Giza-5, and Giza 6) on all evaluated traits. When the interaction between the studied factors was significant according to the P-value (as indicated in ANOVA analysis at the end of Tables 1-4) only these significant interactions were presented in Figures (Figures 1-4). The interaction between the two studied factors was significant only for fresh weight per plant, dry weight per plant, Fe content, number of pods per plant, pod yield, seed yield, protein content, and oil content as presented in Figures 1-4.
Regarding non-treated control, this study was carried out in newly reclaimed sandy soil, and calcium sulfate is recommended for commercial peanut cultivation under these poor conditions which is applied as a control treatment (Ca-1).
In all figures, only 1 decimal place on the y-axis would be sufficient
Re: All figures have been modified to have 1 decimal on the y-axis
Increased content of Fe in plants treated with 50% calcium sulfate + 50% nano-calcium is not discussed. Can it be related to the synthesis of siderophores? In addition, in the discussion section, the effect of sulfate alone is not mentioned thoroughly.
Re: Increased content of Fe in treated plants with Ca-4 and Ca-5 has been discussed (lines 283-291) as well as the effect of sulfate (276-283).
Line 210: Zinc instead of Zink
Re: Done (line 242)
B: boron instead of bron
Re: Done (lines 203, 242)
The last sentence of the discussion (lines 279-282) makes no sense, Ca is not mentioned.
Re: The sentence has been revised and modified as suggested (lines 316-326)
Methods: How many peanut plants did you have in total? How many controls and how many in each group (Ca1-Ca5)?
Re: More details have been added as requested, please see lines 353-354.
What was the commercial source of nano-calcium nitrate? (Producer, country of origin, batch)
Re: More details have been added as suggested please see lines 371-373
Line 290 Please express the conductivity in Siemens.
Re: Done as requested (line 333)
Line 312 potassium sulfate is not K2O
Re: Has been clarified as requested to be “potassium sulfate (K2SO4 contains 48% K2O)” please see line 355
Line 319 is confusing: at a rate 3g l-1 (1440 g/ha), Why two different concentrations? Which is the stock solution?
Re: The foliar solution was prepared at a rate of 3 grams/liter and the hectare requires 480 liters of foliar solution which contained 1440 grams of calcium-nitrate nanoparticles. This has been clarified as requested (line 363)
Line 341: Which methods were used for protein and lipid content determination?
Re: More details have been added as requested (lines 386-388)
Line 360: Why Calcium C in capitals?
Re: Modified to a lowercase letter (line 407)
Writing of references should be uniform, e.g. journal abbreviations and full titles are mixed.
Re: The references have been revised and uniformed
We greatly appreciate the careful review which considerably assisted in improving the manuscript.
Reviewer 2 Report
The selected treatments and comparison on various study variables was excellent. The article is well presented. However, I have some important minor comments are mentioned below.
1. Why you want to compare Nano and Conventional Ca?
2. As you said that Ca is important for pod growth and yield etc, so what you suggest, can your conclusion will be fit for soybean or other other pod plants?
Please also mentioned the logical reason in the introduction and discussion/conclusion section.
3. Please revise line 35-40. Not clear
4. You have presented the abstract is a good format but the values of some important significant variables are missing, Please add it to make a clear idea to readers.
5. I suggest to revise the titles of all tables and figures., such as,
Response of calcium applications (nano and conventional) on.......
6. Write the detail information of statistical significant in the footnote for tables and figures.
7. Why some significant letters are capital and small, provide detail in the footnotes.
Author Response
Dear Editor,
We would like to thank you and the reviewers for the time and efforts devoted to our manuscript entitled “Response of Diverse Peanut Cultivars to Nano and Conventional Calcium Forms Under Alkaline Sandy Soil” (plants-2421890). We have revised the manuscript according to the comments and suggestions pointed out by the reviewers. We have addressed the comments of the reviewers in a point-by-point below in red color; in addition, we have highlighted all the associated changes made to the manuscript using track changes.
Yours sincerely,
Authors
Reviewer 2:
The selected treatments and comparison on various study variables was excellent. The article is well presented. However, I have some important minor comments are mentioned below.
Re: We would like to thank the Reviewer for his time dedicated to our manuscript and his positive assessment of our work.
- Why you want to compare Nano and Conventional Ca?
Re: More information has been added to the introduction as suggested, please see lines 86-94
- As you said that Ca is important for pod growth and yield etc, so what you suggest, can your conclusion will be fit for soybean or other pod plants?. Please also mentioned the logical reason in the introduction and discussion section.
Re: More details have been added as requested in the introduction (65-70) and discussion (245-249)
- Please revise line 35-40. Not clear
Re: Has been revised as requested (lines 35-45)
- You have presented the abstract in a good format, but the values of some important significant variables are missing, Please add it to make a clear idea to readers.
Re: The abstract has been revised and more information has been added as suggested
- I suggest to revise the titles of all tables and figures., such as, Response of calcium applications (nano and conventional) on.......
Re: All titles have been modified as suggested
- Write the detail information of statistical significant in the footnote for tables and figures.
Re: Done as suggested
- Why some significant letters are capital and small, provide detail in the footnotes.
Re: The uppercase letters belong to first season while the lowercase letters to second season. More details have been added to the caption as suggested.
We highly appreciate the careful review which considerably assisted in improving the manuscript.
Round 2
Reviewer 1 Report
The authors have made some progress with their article, but I still recommend correcting the following:
It's still not clear to me, and it might not be to the readers either, in table 1-4 is the mean value of 3 cultivars altogether? This should be specified in the legend. In its current form, it still appears that individual cultivars have been assessed individually.
I know the oxidation number of calcium as a macronutrient (delete from line 66), but what is it in nano-fertilizer, the same?
Protein content was calculated by multiplying the total nitrogen by 5.7. Please add the citation. Lowry or Bradford method would be more suitable.
Minor editing of English language required
Author Response
Dear Editor,
We would like to thank you and the reviewers for the time and efforts devoted to our manuscript entitled “Response of Diverse Peanut Cultivars to Nano and Conventional Calcium Forms Under Alkaline Sandy Soil” (plants-2421890). We have revised the manuscript according to the comments and suggestions pointed out by the reviewers. We have addressed the comments of the reviewers in a point-by-point below in red color; in addition, we have highlighted all the associated changes made to the manuscript using track changes.
Yours sincerely,
Authors
Responses to Reviewers' Comments
Reviewer 1:
The authors have made some progress with their article, but I still recommend correcting the following:
It's still not clear to me, and it might not be to the readers either, in Table 1-4 is the mean value of 3 cultivars altogether? This should be specified in the legend. In its current form, it still appears that individual cultivars have been assessed individually.
Re: The applied experimental design was split plot with randomized peanut cultivars in main plots and calcium fertilizers in subplots. The averages of main effect of studied factors displayed the levels of each factor separately. While the interaction effect is the simultaneous changes in the levels of both factors. Tables 1-4 presented the main effect of studied factors peanut cultivars (Ismalia-1, Giza-5, and Giza 6) and calcium fertilization (Ca1 - Ca5) on all evaluated traits. When the interaction between the studied factors was significant according to the P-value (as indicated in ANOVA analysis at the end of Tables 1-4) only these significant interactions were presented in Figures (Figures 1-4).
More clarifications have been added to the footnotes of Tables 1-4.
I know the oxidation number of calcium as a macronutrient (delete from line 66), but what is it in nano-fertilizer, the same?
Re: Yes, the oxidation number of nano-calcium is the same.
Protein content was calculated by multiplying the total nitrogen by 5.7. Please add the citation. Lowry or Bradford method would be more suitable.
Re: Bradford et al. (1976) have been added to lines 349 and 549
Minor editing of English language required
Re: The manuscript has been carefully revised
We highly appreciate the careful review of both revisions which considerably assisted in improving the manuscript.
Reviewer 2 Report
Article has been significantly improved and accepted for publication.
Author Response
Re: We would like to thank the Reviewer for his time dedicated to our manuscript and his positive assessment of our first revision. We greatly appreciate the careful review which considerably assisted in improving the manuscript.